# Research on Sustainable Management Strategies for the Machine Tool Industry during the COVID-19 Pandemic in Taiwan

**Dyi-Cheng Chen and Tzu-Wen Chen \***

Department of Industrial Education and Technology, National Changhua University of Education, No. 1, Jin-De Road, Changhua 500, Taiwan; dcchen@cc.ncue.edu.tw
**\*** Correspondence: twchen@tcivs.tc.edu.tw

**Abstract:** The machine tool industry is an economically important industry in Taiwan. However, due to the limited natural resources in Taiwan, many of the raw materials required for production must be imported. In 2020, COVID-19, the most serious infectious disease in modern times, broke out across the globe. This has had a great impact on the economic and industrial development of various countries and indirectly affected the development of the machine tool industry. The machine tool industry generally is facing shocks and crises. Therefore, this research article mainly discusses a sustainable operation strategy for the machine tool industry during the COVID-19 epidemic period in Taiwan. Firstly, through the literature on dynamic capability theory (DCT) and expert interviews, the relevant dimensions and criteria are summarized. Then, the fuzzy Delphi method (FDM) and the analytic network process (ANP) are integrated to confirm the relevant dimensions and criteria and to sort the criteria. The five dimensions, in order, are integration ability, learning ability, quality improvement, environmental adaptation, and marketing ability. The ten criteria are diversified learning and innovation ability, integration of multiple sources of knowledge, the ability to learn across departments, the ability to adapt to the external environment, marketing strategy ability, organizational learning ability, integration of resources, improved management efficiency, market research ability, and backward integration. Finally, we put forward business strategies for the ranking results and provide relevant research and industry references.

**Keywords:** analytic network process; COVID-19 pandemic; dynamic capability theory; fuzzy Delphi method; machine tool industry; sustainable management

## 1. Introduction

COVID-19 has become a global pandemic affecting individuals around the world. The development of the coronavirus disease pandemic has led to requirements for social distancing and individual cleanliness measures to secure general wellbeing [1]. Several factors, such as the general fear of COVID-19, quarantine rules, and health recommendations, have altered the activity patterns of people and economic activity [2]. Although scientists have also accelerated research, the characteristics and prevalence of the virus are still full of variables. The actions of various countries and the decision-making barriers of various international organizations not only affect the safety of nations but also the political and economic development of various countries, and they may also affect the evolution of the global situation. The machine tool exhibitions that were expected to be held all over the world have also been postponed or cancelled due to the epidemic situation. This has also made the machine tool industry face an even worse dilemma, related to the epidemic situation and the suspension of the mainland China market. Manufacturers are facing the problem of a significant decline in orders. In addition to relying on governments' relief measures, the problems that machine tool manufacturers must face and overcome include even greater challenges. The response to the changes in the industrial environment

caused by COVID-19 represents a crisis, but it is also a turning point that will make it possible to better store the energy of enterprises for sustainable operations [3]. In addition, performance data in many areas have been revised downward. As a result of this, today's machine tool manufacturers are more committed to reducing operating space in order to reduce job conversion time, reducing raw material transportation costs in order to improve quality, and shortening cycle times in the production of specific products in order to maintain competitiveness. These factors will increase the market's demand for machine tools. After growing in 2018, the global machine tool industry was affected by the US–China trade war; it began to decline sharply in 2019.

In 2020, the world was affected by the COVID-19 pandemic. Taiwan's machine tool exports amounted to USD 2.154 billion, a decrease of 29.7% compared to the same period last year. Japanese machine tool orders decreased by 26.7% compared to the previous year, with an order value of JPY 901.8 billion, which represents a substantial decrease over two consecutive years. In addition, in 2019, the market boom in the United States and the European Union was expected to decline in 2020, and Germany's 2020 annual orders were expected to be reduced by 28%, with production reduced by 30% [4]. Furthermore, it can be observed that the shortages in chip production worldwide have also affected the delivery times for controllers of machine tool, including those from companies such as Fanuc and Mitsubishi. The delivery time has extended from 2.5 months in the past to about 6 months at present. These chips are important. The extension of short-term and long-term delivery periods seems inevitable, as long as the current epidemic situation does not slow down.

The epidemic has almost closed the borders of countries and has stopped many economic activities. In May 2021, in order to address the impact of the COVID-19 pandemic on Taiwan's machine tool industry, the Taiwanese government proposed the allocation of an amount of USD 100 million to purchase Taiwanese-made machine tools and updated machines. It is estimated that 2200 machine tools will be purchased for teaching equipment in 56 colleges and universities and 70 technical high school machinery-related departments so that students of engineering in Taiwan will have better equipment with which to practice. Taiwan's machine tool industry cannot always rely on government subsidies to survive, and the whole world's machine tool industry is affected by the COVID-19 pandemic, the shortage of raw materials, and the shortage of parts [5]. We discuss the sustainable operation strategies for the machine tool industry during the period of industrial stagnation under the impact of the COVID-19 pandemic and after recovery. We examine the corporate value and positioning of the machine tool industry and whether Taiwan's machine tool-related products can meet the needs of global consumers in the international market. Therefore, this article explores and explains the manufacturing factors in the machine tool industry by drawing on previous studies from the literature, and we also consulted experts from the machine tool industry and university professors to sort out the factor dimensions and indicators of the machine tool industry's sustainable operations in response to COVID-19. We used the fuzzy Delphi method (FDM) and an analytic network process (ANP) to screen the important aspects and factor indicators of product manufacturing in the machine tool industry and to explore and develop manufacturing strategies. The purpose of management is to provide references for related industries on issues such as improving product accuracy, shortening manufacturing processes, improving production efficiency, and streamlining the work force, thereby enhancing the competitiveness of industrial operations.

## 2. Literature Review

### 2.1. The Theory of Dynamic Capability

Dynamic capability theory analyses the external environment and defines innovation opportunities, formulates strategies to deal with new opportunities, and finds partners who can complete innovation opportunities, cooperate and compete, offer new competitive advantages, and end existing cooperations [6,7]. Based on the analysis of the external environment, we can discover and define new opportunities, formulate corresponding

strategies, select temporary and discontinuous partners to complete specific strategies, and gain new competitive advantages through cooperation and competition. In addition, we can resolve existing cooperation after the strategy and return to the starting point to find innovation opportunities through continuous innovation to maintain the company's continuous development in a super competitive environment. The rate at which a country's economy grows depends critically on whether its firms can build the capabilities to generate and take advantage of the "windows of opportunity" that exist for innovation and new markets, and whether, over time, they can enhance their capabilities to move into higher value-added activities [7]. The dynamic capability area can be divided into three key concepts: sensitively perceive and identify opportunities, use the flow of resources to capture opportunities to obtain value, and constantly update resources and use them [8,9].

### 2.2. The Theory of Fuzzy Delphi Method

The fuzzy Delphi method (FDM) is a method of research and exploration by experts. It uses the preference judgment of each participant to judge and construct each participant's personal fuzzy preference relationship and uses the group preference relationship to choose the best solution. The project expert forecast method [10–12] is a structured decision support technology. Its purpose is to obtain relatively focused information and opinions through the independent and repeated objective judgments of multiple experts in the process of data collection [11]. In order to improve the deficiencies of the Delphi method, the Delphi method is combined with the fuzzy theory by Murray in 1985, and even the fuzzy integral and the cumulative number of distributions are used to integrate the concept and expert opinions into fuzzy numbers. Thus, the fuzzy Delphi method has established its research methods. This combination method and the traditional Delphi method have the following advantages: (1) it can reduce the number of surveys; (2) it can more completely express the opinions of experts; (3) through the application of fuzzy theory, the knowledge of experts will be more consistent with rationality and demand; and (4) it has more economic efficiency in terms of time and cost [13,14]. Because FDM is a research method for screening a factor, this method can improve the uncertainty and ambiguity of traditional Delphi, so FDM is widely used in function evaluation, planning, and management.

This method has three characteristics, namely: (1) anonymous response, which can reduce questionnaire respondents from being affected by other more influential respondents; (2) iteration and controlled feedback, where the questionnaire for each round depends on the question. Revision of the questionnaire results from the previous round can reduce the disagreement and further gain the consensus of the interviewees. The anonymous feedback process makes group decision-making more accurate, and can eliminate errors and improve accuracy at the same time. (3) The statistical group response can include the opinions of all individual interviewees [12]. The fuzzy Delphi method uses statistical analysis and fuzzy calculation to transform the subjective opinions of experts into quasi-objective data. The fuzzy Delphi method is used for factor selection and comprehensively considers the uncertainty and ambiguity of experts' subjective thinking, so the goals set by the research can be achieved [11,13].

### 2.3. The Theory of Analytic Network Process

The application of the analytic network process (ANP) is based on the analytic hierarchy process (AHP) created by Thomas L. Saaty in 1971, which was developed to deal with complex practical problems [15]. The degree of mutual influence and feedback between dimension and dimension at the problem level can be explored in more depth. Each key decision element is mastered by the hierarchical structure. The name meaning scale is used to make a comparison between the elements, and then a paired comparison matrix obtains the priority vector to show the priority order of the elements. The ANP analysis method is developed, calculated, and analyzed through the AHP analysis method, plus the relationship between dependence and feedback. The AHP analysis method is used to decide based on multiple evaluation criteria and decompose problems into vertical hierarchical

relationships. The ANP method can make up for the lack of correlation assumptions in the analytic hierarchy process [16–18]. ANP can solve non-linear and complex hierarchical relationships, and its decision-making process can reflect real-life phenomena more than AHP. In recent years, the application of ANP has gradually become widespread. ANP is presented as a network, while considering the interdependency between various factors to explain and develop, considering the internal dependencies (criteria and criteria mutual influence, mutual influence between schemes and schemes) and external dependence (interaction between criteria and schemes). Decisions can be reached systematically, and the influence of mutual dependence can be calculated using the super matrix (Supermatrix). The greatest utility of using this method is that the ANP model operation can bring research results closer to decision-making in society, and it also breaks the traditional AHP assumption that decision-making criteria treat each other as independent developments, and the lack of interaction occurs. Satty pointed out that graphics can show the interactive influence of the interdependence between constituent groups and elements, and arrow symbols can represent the relationship and interactive influence between each [15]. This study uses the ANP method to screen the facet factors and indicators of the sustainable operation strategy of the machine tool industry and explores and develops the strategy and management implications of the sustainable operation of the machine tool industry.

*2.4. The Sustainable Management Strategy*

"Sustainability" is an economic development that can meet the needs of this generation without compromising the ability of future generations to meet their needs. As far as enterprises are concerned, this includes issues such as corporate social responsibility, corporate citizenship, improving corporate management of social and environmental affects, and improving the participation of stakeholders, an important role in the processes of social innovation and sustainable development [19]. In recent years, sustainable operation has been paid more attention in order to achieve the goal of sustainable operation of the enterprise. However, when discussing the sustainable operation of enterprises, we should think about its true meaning from a broader perspective [20,21]. The corporate sustainability strategy can be explained by these three aspects. (1) The company attaches importance to R&D and innovation. If an enterprise wants to continue to operate and endure for a long time, it needs to make continuous resource investment in the development of services and products. (2) For research and development, it needs to focus on "technical research" rather than "product development"; research can also extend to protecting of the environment. (3) Contribution of an enterprise to society [22]; if an enterprise can actively contribute its own power and take part in social welfare activities, it will bring a good reputation to the enterprise, prevent it from being affected by negative comments, and achieve the purpose of sustainable operation. In the knowledge-based economy, the technological capability of a firm should form a part of their sustainable development [23,24]. Some companies believe that there is a cost to implementing a sustainable policy, and others believe that, although this is good for the environment and society, it has limited benefits for performance. Nowadays, more and more companies have realized the importance of controlling the company's social and environmental performance. The company implements a corporate strategy that comprehensively considers social, environmental, and economic impacts [25]. The motivation may come from within the company. For example, the management considers sustainability as one of the company's core values, or the management recognizes that sustainability can increase the company's revenue, reduce costs, and create financial value [26]. However, the intrinsic motivation for the sustainability strategy is mostly from external pressure, such as government regulations, market requirements, actions of competitors, or pressure from non-governmental organizations. For senior executives, developing a sustainable strategy has always been an important challenge, but usually, actual implementation is a more daunting challenge. In most cases of successful implementation of sustainability strategies [27,28], the CEO actually takes part and is the promoter of the company's focus on practising sustainability [29]. However, these senior managers often

face a tough challenge: how to improve sustainability performance on three levels at the same time—social, environmental, and financial performance. Business units and factory managers are under pressure to achieve profit goals [30]. Their performance evaluation usually depends on whether they successfully achieve these financial goals. Therefore, the company's strategy, organizational structure, system, performance evaluation, and reward system often fail to properly match, so that the entire company fails to move towards effective implementation of sustainability [31,32]. In addition, companies and personnel often find it difficult to obtain the resources and cannot effectively manage the various motivations for improving social and environmental performance.

## 3. Methodology

### 3.1. Research Procedure

This research mainly discusses research on the sustainable operation strategy of the machine tool industry under the impact of COVID-19 in order to obtain the ranking of the elements and the importance of the index of the sustainable operation strategy of the machine tool industry under the impact of COVID-19 and explore the development and management of its business strategy. The purpose of this research was to use a literature review and expert opinions to draft the important factors of a sustainable business strategy of the machine tool industry and then use FDM and apply the ANP method to obtain to screen out the important key factors of the sustainable business strategy of the machine tool industry. The key dimensions and indicators of sustainable business strategies in the machine tool industry are ranked, as well as research on business strategies and management implications. Based on the research purpose and the results of the literature review, this research proposes the following research hypotheses:

**Hypothesis 1 (H1).** *The business strategy factors of Taiwan's machine tool industry have a positive impact on sustainable management.*

**Hypothesis 2 (H2).** *The important sequence of business strategy factors in Taiwan's machine tool industry has a positive impact on continuous management.*

**Hypothesis 3 (H3).** *The improvement in quality and efficiency of Taiwan's machine tool industry has a positive impact on manufacturing costs and price competitiveness.*

**Hypothesis 4 (H4).** *The innovation of the operator has a positive impact on the company's sustainable development.*

The research process and steps are shown in Figure 1.

### 3.2. Fuzzy Delphi Method, FDM, and Dynamic Capabilities Theory, DCT

This study uses the fuzzy Delphi method to reach an expert consensus on the research topic through group communication. The three major steps of the fuzzy Delphi method are: (1) establish a set of evaluation factors that affect decision-making; (2) collect expert or decision-making group opinions; and (3) calculate the evaluation value of the fuzzy Delphi method. This paper hopes to explore the essential characteristics of dynamic capabilities and their influence on the sustainable management strategy of enterprises in crisis events by tracing the development of dynamic capabilities theory. Warner and Wäger (2019) proposed that the dynamic capability framework can indeed effectively test a powerful indicator of the digital transformation of enterprises, and it remains important in the emerging digital economy [33]. The related research proposed by the aggregate dynamic capabilities theory is as follows; the concept of managerial cognitive capabilities includes three dimensions, sensing, seizing, and reconfiguring, and points out the business operator. The cognitive ability of management will influence strategy changes, form dynamic capabilities, and provide a conceptual basis for the study of dynamic capabilities [34]. According to the theoretical viewpoints and aspects of dynamic capabilities mentioned in the past

literature, dynamic capabilities are the ability to adjust the type of resources with internal or external resources or capabilities in order to cope with changes in market structure and the importance of the company's own cognition for generating dynamic capabilities, reallocating company resources, transforming the new process, strengthening the company's internal core capabilities, and developing inferences about service innovation and organizational performance.

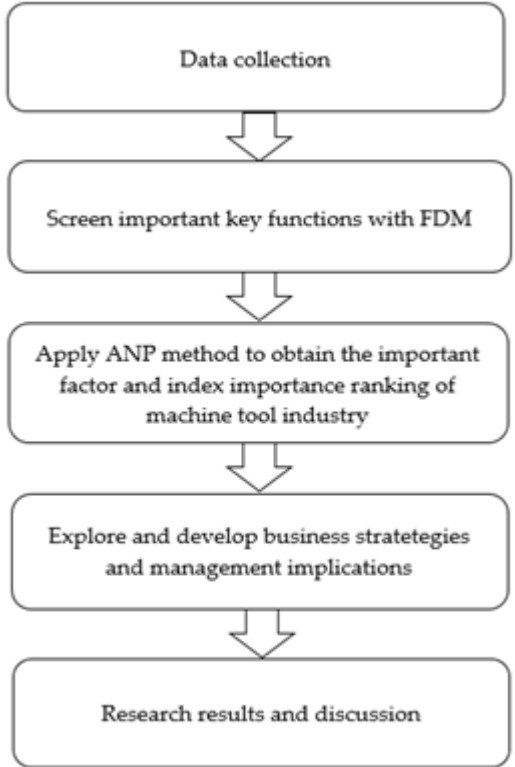

**Figure 1.** Research framework flow chart.

*3.3. Participants*

The analysis steps of ANP are: (1) establish the problem structure, the decision-making problem, and the network hierarchy(2) compare each aspect with the standard in pairs; (3) transform the total influence of the standard into a total importance relationship, normalizing and transposing the matrix at the same time to complete the unweighted matrix of the super matrix; (4) carry out the limited super matrix, and finally, (5) obtain the weight and order of each evaluation standard.

This study uses the fuzzy Delphi method to test machine tool industry experts and publishes a questionnaire on the dimensions and standards of the machine tool industry's sustainable operation during the COVID-19 situation. Eight experts—six general managers and two university professors—completed the fuzzy Delphi expert questionnaire for this research. All the questionnaires were returned, and they were all valid questionnaires. The answering period is from May to August 2021. This period is the most severe period of the COVID-19 attack in Taiwan. Expert background information is shown in Table 1.

**Table 1.** Expert background of FDM.

| Type of Industry | Duty Position | Gender | Educational Background | Seniority |
|---|---|---|---|---|
| Company A | manager | M | Bachelor. Mechanical | 31–35 years |
| Company B | manager | M | Master. Materials engineering | 11–15 years |
| Company C | manager | F | Ph.D. Industrial management | 15–20 years |
| Company D | CEO | M | Master. Mechanical | 11–15 years |
| Company E | manager | M | Master. Precision machinery | 11–15 years |
| Company F | manager | M | Bachelor. Mechanical | 11–15 years |
| University A | professor | M | Ph.D. Educational statistics | 15–20 years |
| University B | professor | M | Ph.D. Mechanical engineering | 15–20 years |

*3.4. Discuss the Sustainable Business Strategy of the Machine Tool Industry under the COVID-19*

As mentioned above, this study uses a combination of qualitative and quantitative methods to conduct questionnaire analysis. Dynamic capability theory was used as the basis of the questionnaire research, and then FDM was used to interview relevant machine tool industry experts for suggestions on the dimensions and guidelines for sustainable operation of the machine tool industry under the COVID-19 situation. After collecting the expert interview questionnaire, the suggestions and feedback were collected and then combined with the ANP to carry out the importance of the important factors of the expert questionnaire and to sort and discuss its management implications.

## 4. Result

This research takes the machine tool industry as the main research object, so that the machine tool industry can develop sustainably.It discusses the important factors and influences of its business strategy and indepth discussion and consultation with experts in the machine industry, and organizes the opinions of the machine tool industry. Indices and procedural methods, FDM, ANP, etc., formulate the best sustainable management strategy for the machine tool industry by ranking factors in order of importance and discussing their management implications.

*4.1. Drawing up Machine Tool Industry Business Strategy Factor Dimensions and Indicators*

Based on the relevant literature and expert opinions from the consulting machinery industry, this study sorted out the factors and indicators of the machine tool industry's business strategy. The factors and indicators of the machine tool industry are shown in Table 2.

*4.2. Fuzzy Delphi Method (FDM) Analysis*

This research consults expert opinions of the machinery industry and the literature to discuss compilation issues and initially drafts the five dimensions and 25 evaluation factors of the machine tool industry's business strategy indicator dimensions and criterion factors. This research invited eight experts to fill in the questionnaire (Table 1). All questionnaires were retrieved, and they were all valid questionnaires. According to the 80/20 rule, we multiplied the total average of 9.62 by 80% to obtain the threshold value of 7.69. With those that were lower are cut off, where all five dimensions are kept (see Table 3), the measurement dimensions have expert consistency and importance. In addition, in the part of the machine tool industry business indicators, this study conducts the evaluation factor analysis of the machine tool industry business strategy indicators. The screening method

uses 90% of the expert consensus value Gi8.65 as the threshold value; the expert consensus threshold value is 7.79. Those who are lower have been deleted, and 22 evaluation criteria are kept, as shown in Table 4.

**Table 2.** Factors and indicators of business strategies.

| | Research dimension | Question of the questionnaire |
|---|---|---|
| Dimension and measurement factors of sustainable operation strategy in the machine tool industry | Integration ability | 1. Integration of resources<br>2. Brand Positioning<br>3. Integration of multiple knowledge<br>4. Manufacturing system integration<br>5. Backward Integration |
| | Learning ability | 1. Creative learning ability<br>2. Adaptability to environmental changes<br>3. Diversified learning and innovation ability<br>4. Ability to learn across departments<br>5. Organizational learning ability |
| | Quality improvement | 1. Changes in the business environment<br>2. Quality analysis and diagnosis capabilities<br>3. Improve management efficiency<br>4. Ability to handle quality abnormalities<br>5. Green productivity ability |
| | Environmental adaptation | 1. Ability to adapt to the external environment<br>2. Ability to adapt to the internal environment<br>3. Ability to adapt to environmental uncertainty<br>4. Ability to adapt to changes in the international environment<br>5. Market environment capability analysis |
| | Marketing ability | 1. Marketing strategy ability<br>2. Sales ability<br>3. Operational capacity<br>4. After-sales service capability<br>5. Market research ability |

**Table 3.** Analysis and selection table of the machine tool industry's business strategy factor dimension.

| Criteria | $C^i$ | | $a^i$ | | $O^i$ | | $M^i$ | | | $M^i$ | $Z^i$ | $M^i$-$Z^i$ | $G^i$ |
|---|---|---|---|---|---|---|---|---|---|---|---|---|---|
| Dimension | Min | Max | Min | Max | Min | Max | $C^i$ | $a^i$ | $O^i$ | | | | |
| Integration ability | 8 | 9 | 9 | 10 | 9 | 10 | 8.42 | 9.42 | 9.87 | 1.44 | 0 | 1.44 | 9.01 |
| Learning ability | 8 | 9 | 9 | 10 | 10 | 10 | 9.19 | 9.99 | 10 | 0.92 | 0 | 0.92 | 10 |
| Quality improvement | 8 | 10 | 9 | 10 | 10 | 10 | 8.78 | 9.78 | 10 | 1.22 | 0 | 1.22 | 10 |
| Environmental adaptation | 8 | 9 | 9 | 10 | 10 | 10 | 8.55 | 9.55 | 10 | 1.34 | 0 | 1.34 | 10 |
| Marketing ability | 8 | 9 | 9 | 10 | 10 | 10 | 8.55 | 9.54 | 9.89 | 1.22 | 0 | 1.22 | 9.01 |
| Total | | | | 5 | | | | | Threshold | | | | 7.69 |

**Table 4.** Analysis and Screening Table of Business Index.

| Criteria | $C^i$ | | $a^i$ | | $O^i$ | | $M^i$ | | | $M^i$ | $Z^i$ | $M^i$-$Z^i$ | $G^i$ |
|---|---|---|---|---|---|---|---|---|---|---|---|---|---|
| Dimension | Min | Max | Min | Max | Min | Max | $C^i$ | $a^i$ | $O^i$ | | | | |
| Integration of resources | 7 | 8 | 8 | 9 | 8 | 10 | 7.64 | 8.53 | 9.52 | 1.88 | −1 | 2.88 | 8.38 |
| Brand Positioning | 7 | 9 | 7 | 8 | 8 | 9 | 7.00 | 7.89 | 9.00 | 2.00 | −2 | 0 | 5 |
| Integration of multiple knowledge | 9 | 10 | 9 | 10 | 10 | 10 | 9.31 | 9.86 | 10.00 | 0.66 | 0 | 0.66 | 10.00 |
| Manufacturing system integration | 7 | 8 | 8 | 9 | 9 | 10 | 7.78 | 8.65 | 9.64 | 1.88 | −1 | 2.88 | 8.26 |
| Backward Integration | 8 | 9 | 8 | 9 | 9 | 10 | 8.65 | 9.65 | 9.87 | 1.22 | 0 | 1.22 | 9.00 |
| Creative learning ability | 8 | 9 | 9 | 10 | 8 | 9 | 8.77 | 9.78 | 9.77 | 1.11 | 0 | 1.11 | 9.00 |
| Adaptability to environmental changes | 9 | 10 | 10 | 10 | 10 | 10 | 9.85 | 10 | 10 | 0.12 | 0 | 0.12 | 10.00 |
| Diversified learning and innovation ability | 7 | 8 | 8 | 9 | 9 | 10 | 7.42 | 8.33 | 93.31 | 1.88 | −1 | 2.88 | 8.65 |
| Ability to learn across departments | 7 | 8 | 8 | 9 | 9 | 9 | 7.31 | 8.20 | 9.20 | 1.88 | −1 | 2.88 | 8.77 |
| Organizational learning ability | 8 | 9 | 9 | 10 | 9 | 10 | 8.31 | 9.31 | 9.76 | 1.44 | 0 | 1.44 | 9.00 |
| Changes in the business environment | 7 | 8 | 8 | 9 | 10 | 10 | 7.64 | 8.65 | 9.65 | 2.00 | −1 | 3.00 | 8.34 |
| Quality analysis and diagnosis capabilities | 7 | 8 | 8 | 9 | 8 | 9 | 7.00 | 8.00 | 8.88 | 1.89 | −1 | 2.89 | 7.00 |
| Improve management efficiency | 7 | 9 | 7 | 9 | 9 | 10 | 7.55 | 8.43 | 9.43 | 1.88 | −1 | 2.88 | 8.51 |
| Ability to handle quality abnormalities | 9 | 10 | 9 | 10 | 9 | 10 | 9.76 | 10 | 10 | 1.45 | 0 | 1.45 | 10.00 |
| Green productivity ability | 8 | 9 | 8 | 9 | 9 | 9 | 8.31 | 9.32 | 9.76 | 1.46 | 0 | 1.46 | 9.00 |
| Ability to adapt to the external environment | 8 | 8 | 7 | 9 | 9 | 10 | 8.25 | 9.25 | 9.31 | 1.57 | −1 | 2.57 | 9.00 |
| Ability to adapt to the internal environment | 6 | 8 | 7 | 8 | 7 | 9 | 7.08 | 8.08 | 9.09 | 2.01 | 0 | 2.01 | 8.00 |
| Ability to adapt to environmental uncertainty | 8 | 9 | 9 | 10 | 10 | 10 | 8.22 | 9.23 | 9.82 | 1.57 | 0 | 1.57 | 9.00 |
| Ability to adapt to changes in the international environment | 9 | 10 | 9 | 10 | 8 | 10 | 9.21 | 10 | 10 | 10 | 0 | 0.25 | 10.00 |
| Market environment capability analysis | 9 | 10 | 10 | 10 | 10 | 10 | 9.55 | 10 | 10 | 0.45 | 0 | 0.45 | 10.00 |
| Marketing strategy ability | 8 | 9 | 9 | 10 | 10 | 9 | 8.22 | 9.22 | 9.76 | 1.55 | 0 | 1.55 | 9.00 |
| sales ability | 6 | 8 | 7 | 8 | 7 | 8 | 6.99 | 7.78 | 8.63 | 1.64 | 1 | 0.64 | 7.62 |
| Operational capacity | 8 | 9 | 8 | 10 | 8 | 9 | 8.24 | 9.24 | 9.25 | 1.50 | 0 | 1.50 | 9.00 |
| After-sales service capability | 9 | 10 | 10 | 10 | 10 | 10 | 9.76 | 10 | 10 | 0.24 | 0 | 0.24 | 10.00 |
| Market research ability | 9 | 10 | 10 | 10 | 10 | 10 | 9.44 | 10 | 10 | 0.56 | 0 | 0.56 | 10.00 |
| Total number of research dimension selected | | | | 25 | | | | | Threshold | | | | 7.79 |

### 4.3. Analytic Network Process (ANP) Analysis

This study uses the fuzzy Delphi expert questionnaire to objectively select and objectively select the dimensions and criteria indicators and then uses the ANP expert questionnaire to discuss the relationship between the dimensions and the criteria of the sustainable operation strategy of the machine tool industry under the COVID-19 situation. We invited eight people. Senior experts selected industry evaluation strategies to understand the mutual influence between dimensions and standards.

In this study, during the COVID-19 situation, the machine tool industry's sustainable business strategy goals are divided into five dimensions, which are represented as A, B, C, D, E: A represents the weight value of the "Integration ability dimension", B represents the weight value of "Learning ability dimension", C represents the weight value of "Quality improvement dimension", D represents the weight value of "Environmental adaptation dimension", and E represents the weight value of "Marketing ability dimension".

The five facets of strategic objectives are displayed through the network-level analysis method; their weights are, respectively, A = 0.225, B = 0.304, C = 0.128, D = 0.184, and E = 0.159, the total weight value is 1, and the formula is $\sum_{5i=1} A_i = 1$

Through the network-level analysis program method, the expert choice tool calculates the weight hierarchy structure diagram of the facet and the criterion and uses MS EXCEL to calculate the facet weight value corresponding to the overall goal and the weight value of all the criterion elements, and it organizes each facet and its criterion elements. The weight values are shown in Table 5.

According to the analysis results in Table 4, the key factors in the machine tool industry business strategy are ranked in the order of "Learning ability", "Integration ability", "Environmental adaptation", "Marketing ability "and" Quality improvement". In addition, the index weighted of the key criteria of the vertical machine tool industry business strategy are weighted and analyzed by the dimension. The final ranking of the top 10 items of importance is, in order of comparison of stations, "Diversified learning and innovation ability", "Integration of multiple knowledge", "Ability to learn across departments", "Ability to adapt to the external environment", "Marketing strategy ability", "Organizational learning ability", "Integration of resources", "Improve management efficiency", "Market research ability", and "Backward Integration".

### 4.4. Analysis of Business Strategy and Management Implication

Affected by COVID-19, countries around the world have implemented domestic blockade control measures to reduce community infections. People can only work remotely. The blockade control measures of countries have not only affected production activities but also the economic and consumer activities of various countries. It also includes the machine tool industry. In order to enable the machine tool industry to operate sustainably, this study selects "Diversified learning and innovation ability", "Integration of multiple knowledge", "Ability to learn across departments", "Ability to adapt to the external environment", "Marketing strategy ability", etc., as business strategy factors and analyses them. We provide specific implementation methods for reference by machine tool industry management personnel. Business strategy and management implications are explained as follows:

(1)    Diversified learning and innovation ability

Innovation must be formed by cooperation between employees and the company. It does not mean that the company can innovate after hiring innovative employees. Therefore, when innovative employees join the company team, the company must adjust its internal policies, support innovative ideas, and put them into action, so that the company can be innovative. Therefore, before finding innovative talents, the pre-work that the company needs to do is very important. A reliable bridge must be established at the same time to create an environment in which all creative content, plans, and work can be easily executed, and spawn countless innovations of substantial social and economic value [35,36].

(2)    Integration of multiple knowledge

Each type of professional knowledge contains various causal relationships or connections between various means and goals. A single knowledge field is usually not enough to deal with problems involving multiple factors and complex causal relationships. Therefore, a combination of different knowledge fields can be considered comprehensive [37]. In combining multiple knowledge fields, the owners of unique fields will use their own professional language to put forward their professional opinions on the same problem or decision. Between various professional fields, on the one hand, there is a complementary and cumulative relationship; they also form various constraints on each other (for example, a workable solution conceived from the perspective of a machine tool designer may be pointed out as infeasible from the perspective of a factory manufacturer). After combining or considering multiple fields, the solution may not only be more creative but also less likely to encounter bottlenecks in the implementation process.

(3)    Ability to learn across departments

Every department and every person in the company has their own role to play, and they also have their own rights to defend. The product department wants to focus on product development, while the sales department wants to focus on product sales, but we all know there are many interdependencies between these two departments, and there are also fuzzy areas in the division of labour. The success of a project often lies in communication. However, communication is not always effective. Effective communication means, "During this communication process, you clearly convey your meaning and achieve your goal". Through the learning of the iliac department, a better communication channel can be established between the various departments so that, in the future, communication can be smoother.

(4)    Ability to adapt to the external environment

From the industry 4.0 era to an industrial society, the rise of technology has intensified industrial competition; with the upgrading of consumer demand and the continuous challenges brought about by industry, the competitive pressure of enterprises is increasing day by day. However, external changes are all faced by society and industry, and the external influence on industry is slow. For enterprises, it is too late, and it is necessary to actively embrace changes [38,39]. The senior executives of an enterprise can guide the enterprise to continuously think about the pressure brought by the environment and guide employees to pay attention to the crisis, which can increase the willingness of employees to change and enhance the value and competitiveness of the enterprise.

(5)    Marketing strategy ability

Marketing is used to create value for customers, and it is procedural and has a certain process. In the process of implementation, many diversified abilities and talents are also needed. In addition to data analysis and data integration, it also requires the integration of all levels to plan the right strategy [40]. Now, companies are paying more and more attention to the fact that everything needs to be supported by data, and it is no longer up to the operator to decide the direction, including international experience and access to relevant information, in which to develop a suitable international marketing strategy [40–42].

**Table 5.** Analysis Table of weighted.

| Dimension | Dimension Weighted | Capability Indicator | Capability Indicator Weighted | Object Weighted | Ranking |
|---|---|---|---|---|---|
| Integration ability | 0.225 | Integration of resources | 0.293 | 0.054 | 7 |
| | | Integration of multiple knowledge | 0.372 | 0.096 | 2 |
| | | Manufacturing system integration | 0.153 | 0.031 | 17 |
| | | Backward Integration | 0.245 | 0.044 | 10 |
| Learning ability | 0.304 | Creative learning ability | 0.169 | 0.032 | 15 |
| | | Adaptability to environmental changes | 0.230 | 0.042 | 11 |
| | | Diversified learning and innovation ability | 0.645 | 0.161 | 1 |
| | | Ability to learn across departments | 0.336 | 0.060 | 3 |
| | | Organizational learning ability | 0.270 | 0.054 | 6 |
| Quality improvement | 0.128 | Changes in the business environment | 0.130 | 0.024 | 21 |
| | | Improve management efficiency | 0.236 | 0.048 | 8 |
| | | Ability to handle quality abnormalities | 0.149 | 0.028 | 19 |
| | | Green productivity ability | 0.152 | 0.028 | 18 |
| Environmental adaptation | 0.184 | Ability to adapt to the external environment | 0.228 | 0.057 | 4 |
| | | Ability to adapt to the internal environment | 0.224 | 0.041 | 12 |
| | | Ability to adapt to environmental uncertainty | 0.171 | 0.033 | 16 |
| | | Ability to adapt to changes in the international environment | 0.184 | 0.031 | 14 |
| | | Market environment capability analysis | 0.149 | 0.021 | 20 |
| Marketing ability | 0.159 | Marketing strategy ability | 0.223 | 0.058 | 5 |
| | | Operational capacity | 0.098 | 0.018 | 22 |
| | | After-sales service capability | 0.179 | 0.036 | 13 |
| | | Market research ability | 0.179 | 0.047 | 9 |

## 5. Discussion

All industries have been affected by the COVID-19 pandemic, which has caused changes in their lives, including various industrial economic activities and school teaching activities. In the future, they must also learn how to coexist with the virus.

Through this research, it is expected that the development of the machine tool industry in the future will be more precise and faster, and manufacturing and processing fields will be broader. In addition, the elements of Industry 4.0 will be added as a more decision application for the future upgrade of the industry.

In terms of research limitations, this research has conducted statistical procedures through data collection, expert interviews, and preliminary analysis of data. The research process strives to be rigorous and still has the following limitations:

(1) The machine tool industry will cause differences in the results of the questionnaire due to the product attributes and the size of the area. This factor will also affect some of the study results of the study.

(2) Due to the limitation of research time and workforce, this research did not conduct further in-depth simulations for industry players, so the results could be more generalized into the research result.

The machine tool industry is a global industry, and the machine tool industry is related to everyone's daily life. This research was conducted in Taiwan, and we look forward to performing further research in Western countries or undertaking homogeneous research to compare the differences of different cultures on the same topic. If there are researchers who require related research in the future, please contact us via email.

## 6. Conclusions

This research uses Taiwan's machine tool industry business strategy to respond to the impact on the machine tool industry during the COVID-19 epidemic. We use the fuzzy Delphi method FDM and the analytical network procedure method ANP to conduct an empirical analysis combined with expert opinions before extracting. The ten important strategic factors, in order of importance, are: "Diversified learning and in-novation ability", "Integration of multiple knowledge", "Ability to learn across departments", "Ability to adapt to the external environment", "Marketing strategy ability", "Organizational learning ability", "Integration of resources", "Improve management efficiency", "Market research ability", and "Backward Integration". The important key factors and research results extracted by this research can provide relevant industry players to improve product quality and improve the quality of human resources. In today's manufacturing industry, because of the increasing shortage of labor and the increasingly sophisticated production equipment, the requirements for product precision, efficiency and quality are increasing. High precision and high efficiency have become the subject of time-consuming attention in precision machining. The project that the machine tool industry attaches importance to is not improving quality and efficiency. Product accuracy and delivery speed are related to customer satisfaction, while quality improvement and efficiency improvement are related to manufacturing costs and price competitiveness in the industry. The continuous operation strategy of the machine tool industry is to respond to changes with changes, replace passivity, and find their own customers. In the future, industry operators and corporate executives must continue to face challenges and innovative breakthroughs in order to make the company sustainable. The sustainable development goals explore the corporate vision and understand the company's impact on society and the environment through impact assessment; then, they plan sustainable competitiveness strategies from the business model diagram and discuss the action plans that can be implemented within the company in the post-epidemic era. They continuously solve social or environmental problems, maximize the living conditions of human beings and the environment, and transform or develop corporate social responsibility into a sustainable enterprise organization. This is also the contribution of this research to sustainable development.

**Author Contributions:** All authors contributed meaningfully to this study. D.-C.C. and T.-W.C.: research topic; data acquisition and analysis; methodology support; original draft preparation; writing review and editing. All authors have read and agreed to the published version of the manuscript.

**Funding:** This research received no external funding.

**Institutional Review Board Statement:** Not applicable.

**Informed Consent Statement:** Not applicable.

**Data Availability Statement:** Not applicable.

**Conflicts of Interest:** The authors declare no conflict of interest.

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
