# Peer review of "Research on Sustainable Management Strategies for the Machine Tool Industry during the COVID-19 Pandemic in Taiwan"

_sustainability, doi:10.3390/su132313449_

Round 1
Reviewer 1 Report
Table 1., Marketing ability - Sales ability - correct
Include future research
Author Response
Responses to the comments made by Reviewer 1
Table 1. Marketing ability - Sales ability – correct.
Reply: We have revised the reviewers' comments, and the revised position is in table2, Thank you reviewer for the suggestion.
Include future research.
Reply: We revised the reviewer's comments, "future research opportunities" into the manuscript, The revised position is line 446-459.

Reviewer 2 Report
- A brief summary
Theme of the paper looks like “hot” one at a glance, as authors “wrapped” core focus on strategic direction development of certain branch of industry into topical issues caused by pandemic situation. The paper does not bring highly innovated information because of the machine tools industry generally faces unpleasant crises not only in Taiwan at present, but throughout the world! Strengths and weaknesses of the paper are not balanced at all, since weaknesses unfortunately clearly prevail! Overall structure of the paper is confusing, literature review is not oriented to main topic and English is rather poor. And additionally, one principal question has occurred: would be research results presented in section 4 distinct from findings of investigations provided in the course of time without pandemic threat? In total, quality of the paper is insufficient.
- Broad comments
Strengths:
- Authors used relevant and suitable methods, especially for data processing.
Weaknesses:
- It is not clear if two questions (lines 68 – 70) are so called research questions.
- Section “Literature review” does not pay attention to the main topic which is sustainable management strategy. Literature review does not provide a state of the art about the main concepts that are discussed in this paper. On the contrary, there is included only information related to methods used by authors during their research. Such information should be incorporated into section 3 “Methodology”. Hence the section two cannot be accepted!
- Clearly defined research hypotheses based on literature review are missing too.
- Lines 29 – 39 are useless as authors describe commonly known information.
- When authors describe the theory of dynamic capability (DCT) as a method used during research works, they unfortunately do not analyze differences between DCT and other relevant methods and approaches to the identification and establishing organization´s strategic direction.
- Last two steps in Figure 1 are the same. Why?
- Information regarding to data gathering is also missing in section 3. Especially: when data were collected, how many experts and respondents were asked for response, who conducted interviews, etc.
- Identifications of research limitations, research gaps as well as opportunities for future research are also completely missing in section 5 “Discussion and conclusion”.
- Text of the paper is badly structured, particularly in sections 2, 3 and 4.
- List of references does not reflect all important resources in area of the main concept that is discussed in this paper: the sustainable management strategy development.
- English is poor, a lot of various mistakes occur in the text.
Author Response
Responses to the comments made by Reviewer 2
It is not clear if two questions (lines 68 – 70) are so called research questions.
Reply: We have revised the reviewers' comments, and the revised position is line 74-88.
Section “Literature review” does not pay attention to the main topic which is sustainable management strategy. Literature review does not provide a state of the art about the main concepts that are discussed in this paper. On the contrary, there is included only information related to methods used by authors during their research ​Such information should be incorporated into section 3 “Methodology”. Hence the section two cannot be accepted!
Reply: We have checked literature review, we have revised the reviewers' comments, and the revised position is line 172-215, Thank you reviewer for the suggestion.
Lines 29 – 39 are useless as authors describe commonly known information.
Reply: We have revised the comments of the editors and added the impact of COVID-19 on the machine tool industry, and the revised position is line 29-52.
When authors describe the theory of dynamic capability (DCT) as a method used during research works, they unfortunately do not analyze differences between DCT and other relevant methods and approaches to the identification and establishing organization´s strategic direction.
Reply: We have revised the reviewers' comments, and the revised position is line 244-258.
Last two steps in Figure 1 are the same. Why?
Reply: We have revised the reviewers' comments, and the revised position is in Figure 1, The reason why the last two steps are the same is an error in editing the manuscript.
Information regarding to data gathering is also missing in section 3. Especially: when data were collected, how many experts and respondents were asked for response, who conducted interviews, etc.
Reply: We have revised the reviewers' comments, and the revised position is in table 1, and line 268-274.
Identifications of research limitations, research gaps as well as opportunities for future research are also completely missing in section 5 “Discussion and conclusion”.
Reply: We revised the reviewer's comments, divided the discussion and conclusion into two parts, and then added the reviewer's suggestions "research limitations” and "future research opportunities" into the manuscript. The revised position is line
446-459.
Text of the paper is badly structured, particularly in sections 2, 3 and 4.
List of references does not reflect all important resources in area of the main concept that is discussed in this paper: the sustainable management strategy development.
Reply: We revised the reviewers' opinions, adjusted the manuscript according to the research framework of the paper, and added support for literature review.
English is poor, a lot of various mistakes occur in the text
Reply: We reviewed the usage and structure of English from a new perspective, and made amendments based on the reviewers’ comments, Thank you reviewer for the suggestion.

Reviewer 3 Report
The paper discusses the sustainable management strategy of the machine tool industry during industrial stagnation and after recovery under the impact of the COVID-19 pandemic in Taiwan. It is a well-written paper, addressing an actual topic. The results are presented clearly and analysed appropriately, and the findings can be useful for decision-makers.
There are a few suggestions for further improvement:
- Discussion and conclusion sections should be separated and extended.
- Please clearly state the key findings and their policy implications.
- Add limitations and directions for further research.
- Given that the authors identify the main issues facing the machine tool industry in Taiwan, the results obtained are specific to this country. Therefore, it would be useful to point to which extent their main findings can be relevant for companies in other countries as well.
Author Response
Responses to the comments made by Reviewer 3
Discussion and conclusion sections should be separated and extended.
Please clearly state the key findings and their policy implications.
Add limitations and directions for further research.
Given that the authors identify the main issues facing the machine tool industry in Taiwan, the results
obtained are specific to this country.
Therefore, it would be useful to point to which extent their main findings can be relevant for companies in other countries as well.
Responses 1:
We revised the reviewer's comments, divided the discussion and conclusion into two parts, and then added the reviewer's suggestions "research limitations” and "future research opportunities" into the manuscript. The revised position is line 446-459. Thank you reviewers for suggestions and comments.

Round 2
Reviewer 2 Report
Clearly defined research hypothese should be included into beginning of section 3. Other comments are revised and can be accepted.
Author Response
Dear Reviewers
Thank you for your letter again dated November 30 2021. We were pleased to know that our work was rated as potentially acceptable for publication in a journal, subject to adequate revision. We thank the reviewers for the time and effort they have put into reviewing the previous version of the manuscript. Their suggestions have enabled us to improve our work. Based on the instructions provided in your letter, we uploaded the file of the revised manuscript. I have a point-by-point response to the comments raised by the reviewers. We hope that the revised manuscript will be accepted for publication in Sustainability.
Sincerely,
Tzu-Wen,Chen
Responses to the comments made by Reviewer 2
Clearly defined research hypothese should be included into beginning of section 3.
Response 1: We have revised the reviewers' comments, and the revised position is line 235-245,Thank you for your comments and suggestions.
